# Investigation on the Slag-Steel Reaction of Mold Fluxes Used for Casting Al-TRIP Steel

**Kaitian Zhang [1,2], Jianhua Liu [2] and Heng Cui [1,***

[1]  Collaborative Innovation Center of Steel Technology, University of Science and Technology Beijing, Beijing 100083, China; zhangkaitianbk@163.com
[2]  Engineering Research Institute, University of Science and Technology Beijing, Beijing 100083, China; liujianhua@metall.ustb.edu.cn
*  Correspondence: cuiheng@ustb.edu.cn; Tel.: +86-136-7123-9796

**Abstract:** The reaction between [Al] in molten steel and ($SiO_2$) in the liquid slag layer was one of the restrictive factors in the quality control for high Al-TRIP steel continuous casting. In this work, the composition and property variations of two slags during a slag-steel reaction were analyzed. Accordingly, the crystalline morphologies of slag were discussed and the solid layer lubrication performance was evaluated by Jackson $\alpha$ factors. In addition, a simple kinetics equilibrium model was established to analyze the factors which affected $SiO_2$ consumption. The results reflected that slag-steel reacted rapidly in the first 20 minutes, resulting in the variation of viscosity and the melting temperature of slags. The slag-steel reaction also affected the crystal morphology significantly. Slag was precipitated as crystals with a higher melting temperature, a higher Jackson $\alpha$ factor, and a rougher boundary with the consumption of $SiO_2$ and the generation of $Al_2O_3$. In other words, although generated $Al_2O_3$ acted as a network modifier to decrease the viscosity of the liquid slag layer adjacent slab shell, the consumption of $SiO_2$ led to the deterioration of the lubrication performance in the solid slag layer adjacent copper, which was detrimental to the quality control for high Al-TRIP steel. Finally, a kinetics equilibrium model indicated that it is possible to reduce a slag-steel reaction by adjusting factors, such as the diffusion coefficient k, $c_{SiO_2}$, $\rho_f$ and $L_f$, during the actual continuous casting process.

**Keywords:** Al-TRIP steel; slag; slag-steel reaction; crystalline morphology; equilibrium model

## 1. Introduction

While the "environmentally friendly, safety, long-life and low-cost" have become an international consensus of the automobile industry, attention for developing advanced automobile strength steel have been paid to high Al-TRIP (Transformation Induced Plasticity), which had a good combination of high strength and high toughness [1–4]. However, 1.35% Al-TRIP steel, currently in industrial production, is a kind of typical peritectic steel with volume shrinkage during peritectic reaction L + $\delta$ → $\gamma$ [5]. As a result, the inhomogeneous growth of the solidification shell increases the incidence risk of the slab surface defects [6]. During the continuous casting process, slag plays an important role in counteracting this phenomenon. It controls the quality of slab through the properties of non-metallic inclusions absorption, lubrication, thermal transmission, etc. Therefore, a system of slag with suitable functions, such as the chemical composition, viscosity, and crystallization morphology, was greatly befitting for a continuous casting process [7,8]. Study on the behavior of slag, especially for the physicochemical characteristics of liquid slag film cling to the slab shell and the crystallization properties of solid slag film attaching to copper, will provide profound guidance for enhancing the slab surface quality of high Al-TRIP steel.

Many investigations on slag used for continuous casting of high Al content steel have been carried out. Conventional slags were mainly based on the $CaO-SiO_2-CaF_2$ system. However, because of Equation (1), which would take place during the continuous casting of high Al steel, the chemical composition of slag would change rapidly into an obvious increase in $Al_2O_3$ and decrease in $SiO_2$, leading to the deterioration of slag physicochemical properties and casting [9].

$$4[Al] + 3(SiO_2) \rightarrow 2(Al_2O_3) + 3[Si] \tag{1}$$

Yu et al. [10] tried to use a high $SiO_2$ with a low basicity flux to counteract the increase in basicity caused by the $(SiO_2)$-[Al] reaction, and the casting experiments showed that the slab had a good surface quality. Zhang et al. [11] also studied the variation in viscosity and crystallization properties by adjusting $Al_2O_3/SiO_2$, however, the improvement effect on continuous casting was not significant. Some researchers have tried to suppress the deterioration in the performance of slag by adding fluxing agents [12,13]. Park et al. [13] studied the addition of $CaF_2$ that was beneficial to the reduction of the viscosity of high Al steel slag, but because of the high basicity of $Na_2O$, when the content of $CaF_2$ exceeds 8%, the viscosity-reducing effect was weakened significantly.

Primarily, it was the drastic slag-steel reaction during the continuous casting process that led to slag performance deterioration. Many attempts based on conventional slag still find it difficult to avoid the slag-steel reaction effectively. Therefore, some researchers began to explore the study of non-reactive or weakly reactive slag [6,14–21]. Among them, Wang et al. [16,17] studied the effect of various oxide additions on the crystallization behavior and heat transfer properties of the $CaO-Al_2O_3$ system continuous casting slag and provided some guidelines for the design of a reasonable $CaO-Al_2O_3$ system slag. Cho et al. [18] designed the $CaO-Al_2O_3$ based slag for continuous casting of 1.45% Al steel and found that the reactivity of Equation (1) was reduced significantly, resulting in a certain improvement of the slab quality. Seo et al. [19,20] focused on the lubrication of a solid layer of slag used for high Al-AHSS (Advanced High Strength Steel) steel and proposed to control the crystalline morphology by Jackson $\alpha$ factors. It was found that the crystallization at the slag film was too strong to affect the lubrication, which ultimately led to defects in the slab.

In summary, the current design schemes of slag still could not continuously cast high Al-TRIP steel perfectly caused by exorbitant $Al_2O_3$ and free of $SiO_2$ content in slag after the reaction. And the additions of fluxing agents, such as $Li_2O$ and $Na_2O$, would also lead to cost problems for industrial production. As a result, Cho et al. [21] proposed a slag feeding technology to improve the castability and surface quality of a continuous cast TWIP (Twinning-induced plasticity) steel. Therefore, low-reactive slag, rather than non-reactive slag, might be a suitable and economic method for the continuous casting of high Al-TRIP steel. The influence of consuming $SiO_2$ during continuous cast Al-TRIP steel slab still requires further study.

Based on the investigation of the composition variation of $CaO-SiO_2$ system slag in an actual Al-TRIP steel continuous casting process, a kind of low-reactive slag based on $CaO-SiO_2-Al_2O_3$ system was adopted in this work. The initial low-reactive slag and corresponding slag samples after 10 min, 20 min, and 120 min of slag-steel high-temperature reaction were analyzed to determine their compositions and properties variation. Next, the crystalline morphologies were investigated by BSE (Back Scattering Electron) and Jackson $\alpha$ factors were adopted to evaluate solid layer lubrication performance. Finally, a kinetics equilibrium model of $(SiO_2)$ in liquid slag was established to analyze the factors that affected $SiO_2$ consumption.

## 2. Experiments

### 2.1. Materials Preparation

The chemical composition of high Al-TRIP steel is listed in Table 1, samples were cut with a weight of 300 g from an industrial slab. The initial compositions of two types of industrial slag were tested by XRF (X-Ray Fluorescence) shown in Table 2. Among them, *S/A* was the content ratio of $SiO_2$ and

$Al_2O_3$, which represented the reactivity of slag samples; $R$ was the basicity of slag samples, which was represented by the content ratio of CaO and $SiO_2$; η represented the viscosity of slag samples at 1300 °C; and $T_m$ represented the melting temperature of slag samples. It is worth mentioning that sample A was used as the control group in the actual Al-TRIP steel continuous casting production. And sample B was used to high-temperature static balance experiment.

**Table 1.** Main chemical composition of Al-TRIP (Transformation Induced Plasticity) steel, wt.%.

| Fe | C | Si | Mn | P | S | Alt | N |
|----|----|----|----|----|----|----|----|
| bal. | 0.16 | 0.16 | 1.49 | 0.008 | 0.001 | 1.35 | 0.0016 |

**Table 2.** Main chemical compositions and properties of initial slag samples, wt.%.

| Samples | CaO | $SiO_2$ | $Al_2O_3$ | $CaF_2$ | BaO | $Na_2O$ | S/A | R(C/S) | η [1] (Pa·S) | $T_m$ [2] (°C) |
|---------|------|---------|-----------|---------|------|---------|------|--------|--------------|----------------|
| A | 30.57 | 27.31 | 3.03 | 17.26 | 5.65 | 16.17 | 9.02 | 0.147 | 0.087 | 841.88 |
| B | 29.50 | 20.90 | 16.16 | 18.72 | 10.93 | 3.79 | 1.29 | 0.147 | 0.154 | 1109.35 |

[1,2] Calculated by Factsage software 7.2, and η were calculated at 1300 °C.

### 2.2. Experimental Method

Figure 1 shows the schematic diagram of the high-temperature static balance experimental apparatus. 300 g steel was put into Si-Mo high-temperature furnace with zirconia crucible. The steel was heated to 1550 °C by 10 °C/min and then stayed the temperature with argon at the rate of 5 L/min to prevent the oxidation of steel. To simulate the situation of a continuous addition of slag in the industry casting process, excess 60 g pre-melted slag was added on the surface of molten steel and taken out of the furnace after continuing heat preservation for 10 min, 20 min, and 120 min. Finally, after crushing and grinding, the slag chemical compositions were measured by XRF. The viscosities of 1300 °C and melting temperature for samples were calculated by Factsage software 7.2 (GTT-Technologies, Aachen, Germany). In addition, part of the slag sample was embedded in the epoxy resin and sprayed with gold, and then, the microstructure and crystal composition of slag sample B during the high-temperature static experiment were observed by Scanning Electron microscopy (Zeiss, Heidenheim, Germany) in Back Scattering Electron mode.

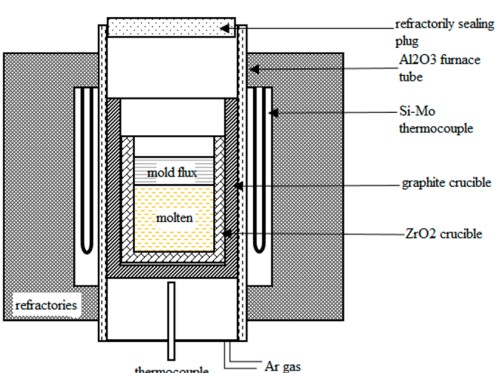

**Figure 1.** Schematic of the experimental apparatus.

## 3. Results

### 3.1. Compositions Variation of Slag during the Slag-Steel Reaction

The main compositions of slag sample A and B in different reaction time (10 min, 20 min, and 120 min) with 1.35% Al-TRIP steel are shown in Figure 2. In the first 10 min, the content of $SiO_2$ decreased sharply while the content of $Al_2O_3$ increased sharply. This means that because of the strong reaction driving force, which resulted in a low content of $Al_2O_3$ and high content of $SiO_2$ in initial

slag samples, both sample A and B reacted rapidly with steel. In the following 10 min, the reaction (1) continued, however, the reactive rates of both samples gradually calmed down by the reason of the actual content ratio of $Al_2O_3/SiO_2$ in slag being increased. In other words, the reaction driving force was weakened. After 20 min, there was no obvious change in the tendency of compositions for sample A and sample B, indicating that the reaction had reached a kinetics equilibrium, and the equilibrium compositions were nearly 20 min after the beginning of the slag-steel reaction. Eventually, there was only 5.69% $SiO_2$ content in slag sample B and its composition had moved from the $CaO-SiO_2-Al_2O_3$ system to $CaO-Al_2O_3$ system. On the other hand, the final content of $SiO_2$ was 19.84% in sample A, which kept sample A in the $CaO-SiO_2-Al_2O_3$ system. In addition, because of the evaporation, no $Na_2O$ was detected in sample B after the reaction.

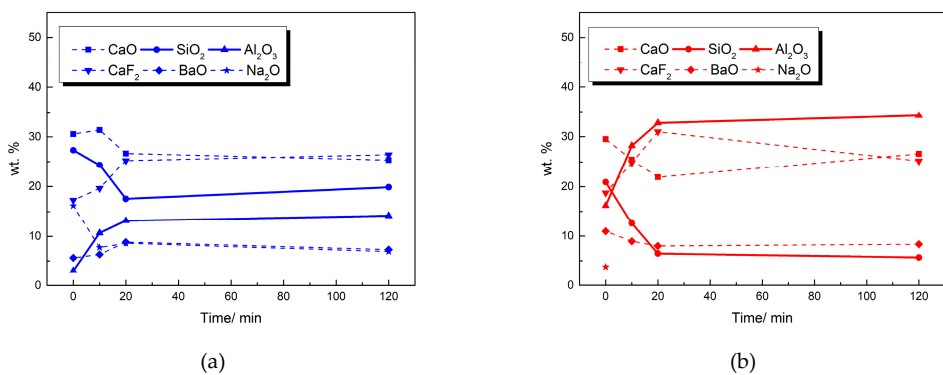

**Figure 2.** Variation of main compositions for slag sample: (**a**) Sample A; (**b**) sample B.

### 3.2. Properties Variation of Slag during the Slag-Steel Reaction

The viscosity at 1300 °C and melting temperature of slag samples before and after the reaction were calculated by Factsage software, shown in Figure 3. It was apparent that the melting temperature increased significantly and then gradually stabilized during the reaction. This was almost consistent with the composition of $Al_2O_3$ with a high melting temperature. The viscosity increased rapidly in the first 10 min of the reaction, and then decreased and gradually stabilized. In general, the variation of the composition and properties of sample A were consistent with that of sample B, indicating that the slag-steel reaction during the actual continuous casting process can be simulated through the high-temperature static balance experiment, so as to analyze the influence of the change of $(SiO_2)/(Al_2O_3)$ content on the properties of slag.

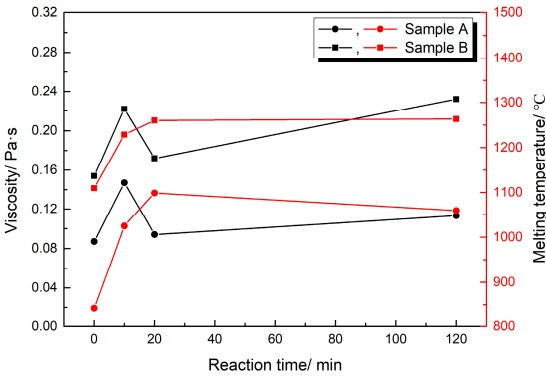

**Figure 3.** Variation of viscosity and melting temperature for slag sample A and B.

### 3.3. Revolution of Crystalline Morphology for Slag during the Slag-Steel Reaction

Figure 4 was the morphology of the original slag sample B. The distribution of slag components for industrial slag was mainly composed of three parts unevenly. Part I was only blocked $Al_2O_3$ with a

high melting temperature. It precipitated prematurely during the continuous casting process, and it performed as amphoteric oxide depending on the basicity of slag. Part II was mainly composed of C, which was a matrix of slag and played a role in assisting melting. Part III was more complex, the main compositions were P1 ($Al_2O_3 \cdot 4SiO_2$), P2 ($2CaSiO_3 \cdot CaAl_2O_4 \cdot BaO$) and a small amount of P3 ($CaF_2$). Among them, P1 existed alone, while P3 existed around P2.

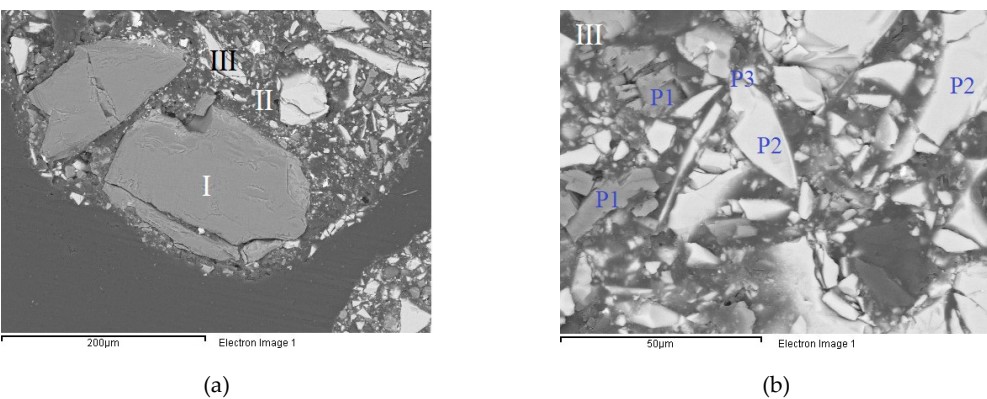

(a)  (b)

**Figure 4.** Morphology of original slag: (**a**) full view; (**b**) details of Part III.

The crystal morphology for slag sample B after the slag-steel reaction experiment (10 min, 20 min and 120 min respectively) are shown in Figure 5. The significant difference in the microstructure of the crystal morphology of the slag with different extent of slag-steel can be clearly observed. As shown in Figure 5a, after the slag-steel reaction lasted for 10 min, the slag matrix was smooth, with a dendritic crystalline phase on the surface but without large crystalline phase. After the slag-steel reacted for 20 min, as shown in Figure 5b, the slag became rough, and there were few large-size planar granular crystalline phases with smooth boundaries on the surface. After the reaction lasted for 120 min, as shown in Figure 5c, the crystalline phase size further increased and the boundary was rougher.

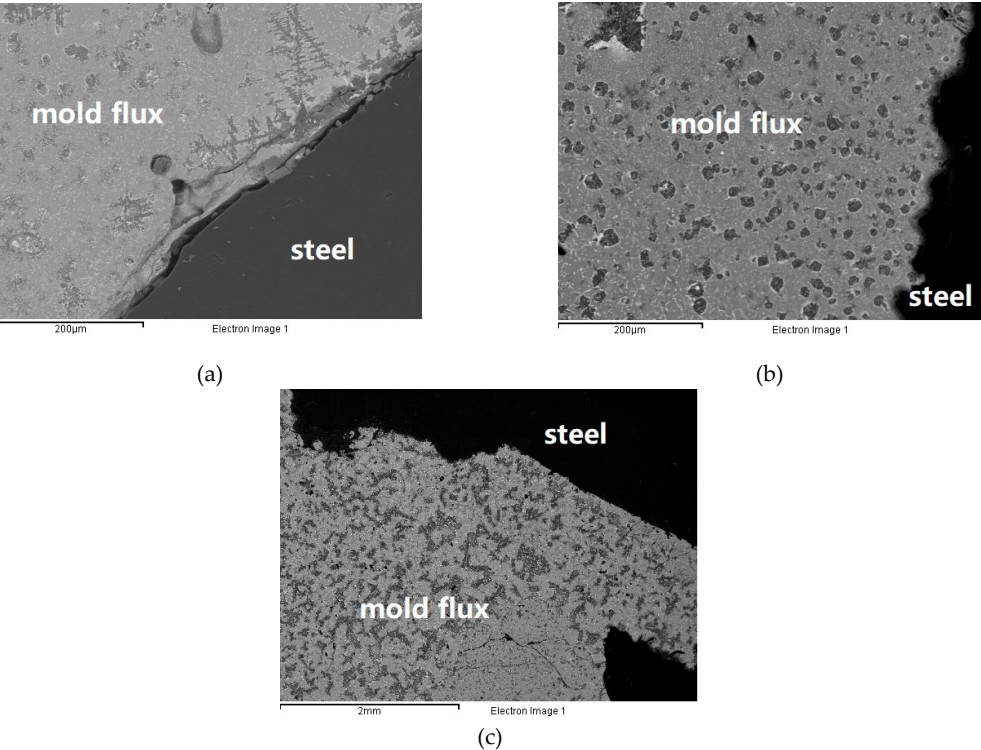

(a)  (b)

(c)

**Figure 5.** Crystals morphology of slag after reaction: (**a**) 10 min; (**b**) 20 min; (**c**) 120 min.

Specifically, Figure 6 shows the details of the slag morphology after the slag-steel reaction for 10 min. The dendritic crystalline phase was $CaF_2$, which was a benefit to lubrication. The matrix of slag crystallized uniformly along the $CaF_2$ dendritic, and the composition was $3CaO \cdot 3SiO_2 \cdot 2Al_2O_3 \cdot BaO$. No cuspidine ($3CaO \cdot 2SiO_2 \cdot CaF_2$) with high melting temperature was observed, indicating the slag had good performance on the uniform thermal transmission and melting characteristic so far.

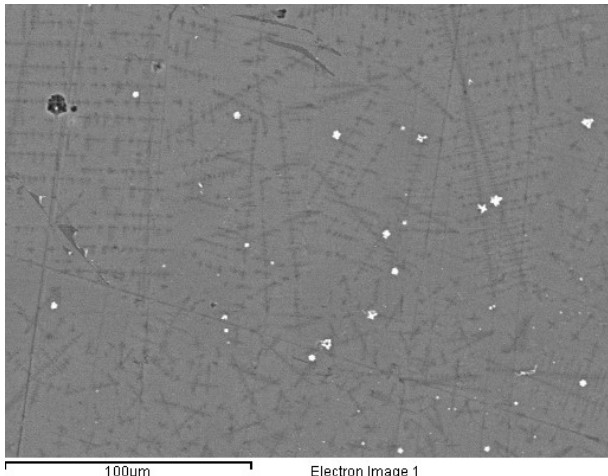

**Figure 6.** Details of crystals morphology after reaction lasted for 10 min.

After the reaction lasted for 20 min, as shown in Figure 7, the morphology of $CaF_2$ (P1) was a granular structure with a smooth boundary, which was favorable for the lubrication performance. The slag crystallized with P1 as the core, and is surrounded by P2 ($CaAl_2O_4$) and few $CaF_2$ dendrites. The main component of a boundary (P3) was aluminosilicates of Ca and Ba, and Ba had a tendency to make up for the formation of fluoride by Ca. The matrix of the slag was $CaAl_2O_4$ and a small amount of $CaSiO_3$, resulting from the large consumption of $SiO_2$.

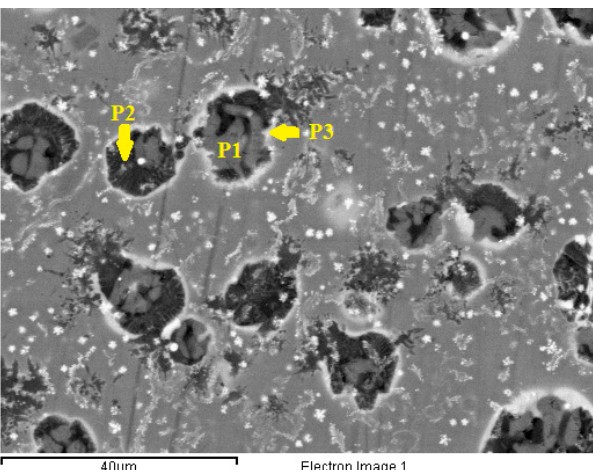

**Figure 7.** Details of crystals morphology after reaction lasted for 20 min.

As the reaction progressed to 120 min (Figure 8), the morphology was similar to that in Figure 6. However, the distribution was dense and uneven with the larger size and sharp boundary, leading to an unfavorable lubrication performance. With the growth of the $CaF_2$ boundary, the surrounding Ba was constantly absorbed into the core, forming $BaF_2$ and Ba-containing calcium aluminate (P2) eventually. The dendritic $CaF_2$ (P3) still existed between the $CaF_2$ core and the boundary. Compared with Figures 6 and 7, the crystal composition in Figure 8 was more complicated. There were Ba-containing $CaAl_2O_4$ and $CaAl_4O_7$ with high melting temperature.

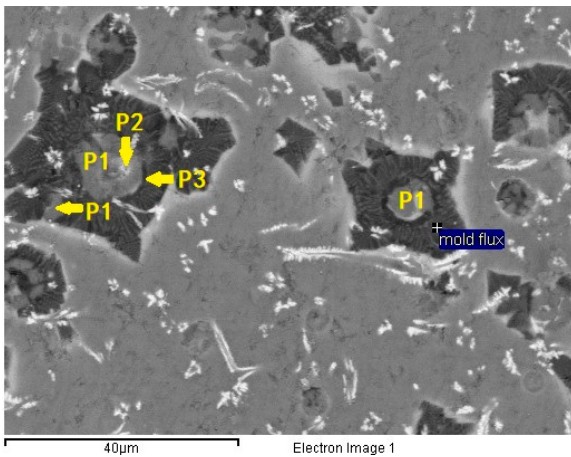

**Figure 8.** Details of crystals morphology after reaction lasted for 120 min.

In addition, Figure 9 was the crystalline morphology at the slag-steel interface after reaction for 120 min. The crystal was dominated by gehlenite and alumina with less fluoride. There were P1 ((Ba, Ca)$F_2$), P2 ($Ca_2SiO_4$), P3 ($3CaAl_4O_7 \cdot BaO$), and P4 ($Al_2O_3$). Since the $Ba^{2+}$ electrostatic potential was smaller than $Ca^{2+}$, P1 was mainly composed of $BaF_2$. P2 was $2CaO \cdot SiO_2$, which was converted from wollastonite ($CaSiO_3$) with a large consumption of $SiO_2$. In addition, the duration of the slag-steel reaction was 120 min, which was close to the reaction equilibrium. Considering the kinetic conditions in the experimental furnace, the $Al_2O_3$ generated by the reaction was concentrated at the slag-steel interface and difficult to diffuse. Therefore, amounts of large-size crystals P3 and P4 could be observed, which had a significant effect on the heat transfer performance of slag-steel interface, and further affected the melting and consumption of slag.

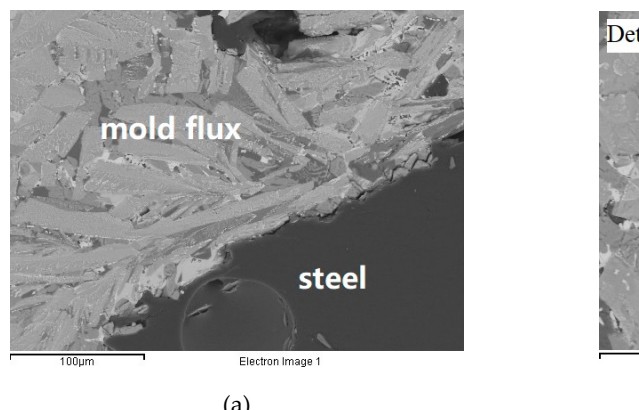

(a)

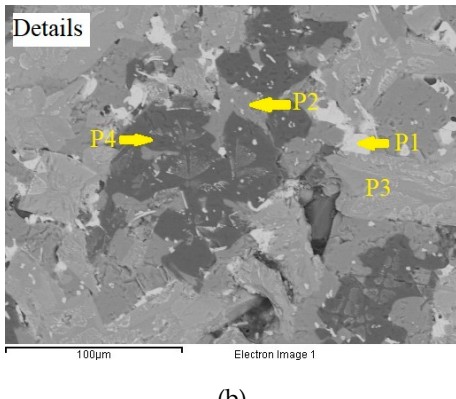

(b)

**Figure 9.** Morphology of original slag: (**a**) full view; (**b**) details of Part III.

## 4. Discussion

### 4.1. The Relationship between the Properties and Compositions Variation of Slag

In an alkalescent melting system, according to previous researchers [16,22], the existential form of $Al_2O_3$ was $[AlO_4]^{5-}$ tetrahedron, which could incorporate into $[SiO_4]^{4-}$ tetrahedral units to act as the network former, resulting in the increase of viscosity. With the content of ($CaO$)/($Al_2O_3$) having decreased, $Al_2O_3$ existed in the form of $[AlO_6]^{9-}$ octahedral, which could act as a network modifier, leading to a decrease of viscosity. Therefore, the viscosity of both slags was not decreased monotonically, it fluctuated slightly in the later period of reaction.

In summary, there were obvious differences in the crystalline phase in each part of the slag after the slag-steel reaction, mainly reflected in the content of $SiO_2$. Different crystalline phase performed

different lubrication effect. The Jackson $\alpha$ factor was adopted in this work to evaluate the roughness of the crystalline phase. The crystals tended to be more faceted and anisotropic when their Jackson $\alpha$ factor increased [20]. The main crystalline phases and their corresponding melting temperature and Jackson $\alpha$ factors were shown in Table 3. As the reaction progressed, the content of $SiO_2$ was gradually reduced while $Al_2O_3$ increased significantly, resulting in a gradual replacement of $CaSiO_3$ and $CaAl_2O_4$ (with lower Jackson $\alpha$ factors) by $CaAl_4O_7$ (with higher Jackson $\alpha$ factor). From the perspective of crystallography, crystals with higher Jackson $\alpha$ factor consisted of crystallographic planes with different orientations, which were distributed at different angles to the heat dissipation direction. This was favorable for the formation of crystals with regular geometrical shapes, and thus the anisotropy of crystals was increased, which resulted in the decrease of lubrication performance in solid slag layer adjacent to the copper. In general, although generated $Al_2O_3$ acted as network modifier to decrease the viscosity of liquid slag layer adjacent slab shell, the consumption of $SiO_2$ led to the deterioration of lubrication performance in solid slag layer adjacent copper, which was detrimental to the quality control for high Al-TRIP steel.

**Table 3.** Typical crystals of slag sample B and corresponding Jackson $\alpha$ factors, wt.%.

| Time | Crystal Phase | Melting Temperature | Jackson $\alpha$ Factors [1] |
|---|---|---|---|
| 10 min | $CaF_2$ (dendritic) | 1418 °C | 2.11 |
| | $CaSiO_3$ | 1540 °C | 3.7 |
| 20 min | $CaF_2$ (faceted) | 1418 °C | 2.11 |
| | $CaSiO_3$ (less) | 1540 °C | 3.7 |
| | $CaAl_2O_4$ | 1604 °C | 3.51 |
| 120 min | $CaF_2$ (faceted) | 1418 °C | 2.11 |
| | $CaAl_2O_4$ | 1604 °C | 3.51 |
| | $CaAl_4O_7$ | 1765 °C | 7.57 |
| 120 min (slag-steel interface) | $CaF_2$ (faceted) | 1418 °C | 2.11 |
| | $CaAl_4O_7$ | 1765 °C | 7.57 |
| | $Al_2O_3$ | 2054 °C | 6.12 |

[1] Based on reference [20].

### 4.2. Kinetics Equilibrium Model of Slag during the Reaction

Non-linear fitting of $SiO_2$ content in both slags through Origin software is shown in Figure 10. These content variations had good agreement with function $c_t - c_0 = a \times [1 - \exp(b \times t)]$. From the perspective of mathematics, coefficient $a$ influenced the limit of the function, and $b$ influenced the inflection time of the function.

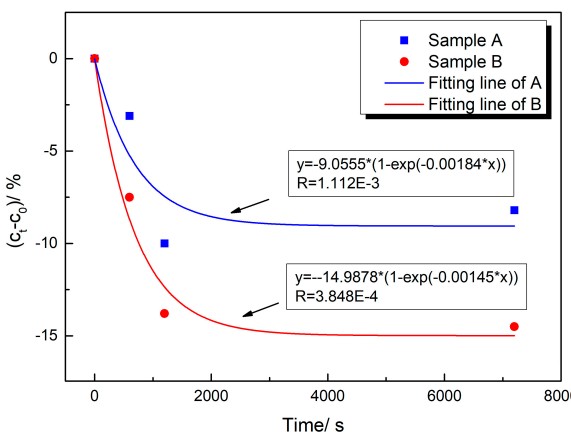

**Figure 10.** Details of crystals morphology after reaction lasted for 120 min.

Therefore, the reactivity of the slag was closely related to coefficient *a* and *b*. Theoretically, during the casting of 1.35% Al-TRIP steel, a certain content of SiO$_2$ in the powder slag was consumed because of reaction (1), and the rest of SiO$_2$, accompanied by molten slag, was flowed into the gap between solidification shell and copper. Therefore, the kinetics equilibrium balance of SiO$_2$ content in the liquid slag layer could be expressed as Figure 11.

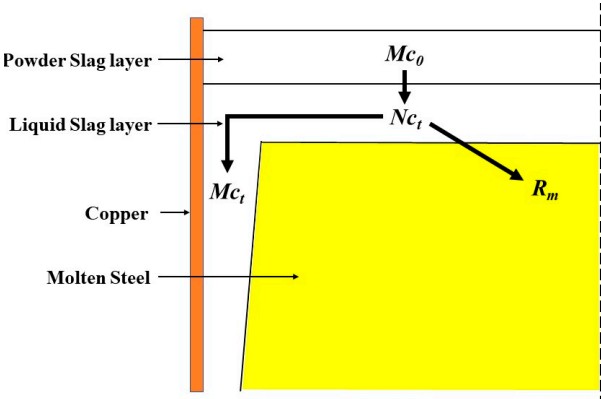

**Figure 11.** Kinetics equilibrium balance of SiO$_2$ content in liquid slag layer.

The $c_0$ was the content of SiO$_2$ in the initial powder slag. At time *t*, the SiO$_2$ content in the liquid slag decreased to $c_t$. Thus, the SiO$_2$ content flow into the gap between copper and slab shell at this time was also $c_t$. The Rm was the consumption of SiO$_2$ caused by the slag-steel interface reaction. Assuming that the supplementary of liquid slag from powder slag layer was equal to consumed slag from liquid slag layer during the continuous casting, and the SiO$_2$ content in the liquid slag layer was uniform at time *t*. Thus, the mass balance of SiO$_2$ in the liquid slag layer could be expressed as follows:

$$\frac{d[\text{SiO}_2]_t}{dt} = \frac{M[\text{SiO}_2]_0 - 100R_m - M[\text{SiO}_2]_t}{N} \tag{2}$$

where, $[\text{SiO}_2]_t$ was SiO$_2$ content in the liquid slag layer at time *t*, %; $[\text{SiO}_2]_0$ was SiO$_2$ content in initial powder slag, %; *M* was the consumed rate of liquid slag, kg/s; *N* was the mass of liquid slag layer, kg; $R_m$ was the SiO$_2$ consumption rate from slag-steel interface reaction, kg/s; and *t* was time, s.

The slag-steel reaction consumed SiO$_2$ and produced Al$_2$O$_3$. In a constant temperature condition, the SiO$_2$ consumption rate $R_m$ was associated with [Al] content in steel and the initial SiO$_2$ content in powder slag. If the reaction was adequate during the experiments, $R_m$ could be expressed as follows:

$$R_m = Akc_{\text{Al}}^m c_0^n \tag{3}$$

where, *A* was reaction area, cm$^2$; *k* was diffusion coefficient related to flow velocity and viscosity. Thus, the Equation (1) could be rewritten as Equation (4) by an integral method.

$$c_t - c_0 = -\frac{100}{\rho_s v_s Q} \times Akc_{\text{Al}}^m c_0^n \times \left[1 - \exp\left(\frac{\rho_s v_s Q}{\rho_f L_s}\right)\right] \tag{4}$$

where, $c_{\text{Al}}$ was the [Al] content in steel, %; $\rho_s$ was molten steel density, g/cm$^3$; $\rho_f$ was slag density, g/cm$^3$; *Q* was slag consumption for steel, g/g; $V_s$ was equivalent casting speed, cm/s; $L_f$ was thickness of liquid slag layer, cm. Combined with Figure 10, the coefficients a and b could be written as follows:

$$a = -\frac{100}{\rho_s V_s Q} \times (k(V_s c_{\text{Al}})^m c_{\text{SiO2}}^n \tag{5}$$

$$b = -\frac{\rho_s V_s Q}{\rho_f L_f} \tag{6}$$

In this work, the temperature, $c_{Al}$, $\rho_s$, $Q$, $L_f$ and $V_s$ could be considered as constant, while $\rho_f$, $L_f$, $c_{SiO_2}$ and $k$ were different. Therefore, from the perspective of the slag-steel reactivity, $k$ and $c_{SiO_2}$ influenced the limit $SiO_2$ consumption of the reaction and $\rho_f$ and $L_f$ influenced the $SiO_2$ consumption rate of the reaction.

## 5. Conclusions

This work investigated the variations of slag components, properties, and crystalline phases during a slag-steel reaction. The following conclusions can be made:

(1)   The components and properties variated rapidly in the first 20 min of the slag-steel reaction and then stabilized gradually. Specifically, slag moved from $CaO\text{-}SiO_2\text{-}Al_2O_3$ system to $CaO\text{-}Al_2O_3$ system, and the basicity of the slag was increased significantly because of the large consumption of $SiO_2$. However, the generated $Al_2O_3$ also acted as network modifier in the slag, resulting in a limited decrease in viscosity of liquid slag layer adjacent to the slab. In addition, a small amount of $SiO_2$ still existed in the slag at the reaction equilibrium point.

(2)   The components variation during the slag-steel reaction also affected the crystal morphology significantly.   Accompanied by the consumption of $SiO_2$ and the generation of $Al_2O_3$, the crystallization phase of the slag tended to be dendritic $CaF_2$ → faceted $CaF_2$ and $CaSiO_3$ → $CaAl_2O_4$ → $CaAl_4O_7$. These phenomena indicated that as the reaction progressed, the slag used for 1.35% Al-TRIP steel was precipitated as crystals with a higher melting temperature, higher Jackson $\alpha$ factor, and rougher boundary. As a result, the anisotropy of crystals was increased and the lubrication performance in the solid slag layer adjacent the copper was deteriorated.

(3)   According to the nonlinear fitting of the slag composition variation during the reaction, it was found that the slag-steel reaction existed the consumption limit of $SiO_2$. A kinetics equilibrium mold of slag was also derived, which indicated that the perspective of slag, the diffusion coefficient $k$ and $c_{SiO_2}$ affected the limit $SiO_2$ consumption of the reaction, and $\rho_f$ and $L_f$ affected the $SiO_2$ consumption rate. Therefore, it is possible to reduce the slag-steel reaction by adjusting these parameters during the actual continuous casting process, and it can provide a new idea for the research of slag used for high Al-TRIP steel.

**Author Contributions:** Investigation, K.Z. and H.C.; project administration, H.C. and J.L.; writing—original draft, K.Z.; writing—review & editing, K.Z. and H.C.

**Funding:** This research was funded by the National Natural Science Foundation of China (No. U1860106).

**Conflicts of Interest:** The authors declare no competing interests.

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
