# Peer review of "Investigation on the Slag-Steel Reaction of Mold Fluxes Used for Casting Al-TRIP Steel"

_metals, doi:10.3390/met9040398_

Reviewer 1 Report

Please change “et al” to “et al.” in the text.

Wang et al [16,17] studied…

should be:

Xiao et al. [16] and Lu et al. [17] studied…

Please make an analog modification for:

Seo et al. [19,20] focused…

…high Al-AHSS steel and…

should be:

…high Al-AHSS (Advanced High Strength Steel) and…

Please check and modify, if necessary.

…continuous cast TWIP steel.

should be:

…continuous cast TWIP (Twinning-induced plasticity) steel.

Please check and modify, if necessary.

…10minutes, 20minutes, and 120minutes of slag-steel…

should be:

…10 minutes, 20 minutes, and 120 minutes of slag-steel…

Change anywhere in the text.

Table 1. Main chemical compositions of Al-TRIP steel, wt%.

If authors are speaking about “Main chemical compositions”, they have to add the “iron content” of the Al-TRIP steel.

Table 2. Main chemical compositions of industrial slag samples, wt%

It seems that the total of oxides contents + F­ content surplus 100%. In addition, if calcium bounded to F­ (CaF2) is added, the sum of all the constituents of the slag will be well over 100%.

Hence, Please check your data and change if necessary.

Please explain the below abbreviations used in Table 2.

S/A

R(C/S)

η1)/(Pa·S)

Tm2)/

Figure 2.

Due to  many information incorporated in Figure 2, it is difficult for the reader to understand this Figure. To be clearer, the experimental results of samples A and B should be put in the form of the two separate Figures, side by side (as Figure 2(a) and Figure 2 (b)).

Furthermore, a rough examination of this Figure showed the initial contents of Na2O for sample A and B are not correct. Compare these data with those given in Table 2. Please check and modify.

3.3 Revolution of crystalline morphology for slag during the slag-steel reaction

should be:

3.3 Revolution of Crystalline Morphology for Slag during the Slag-Steel Reaction

The authors stated:

“Figure 4 was the morphology of the original slag sample B. The distribution of slag components for industrial slag was mainly composed of 3 parts unevenly. Part I was only blocked Al2O3 with a high melting temperature. It would precipitate prematurely during the continuous casting process, and it would perform as amphoteric oxide depending on the basicity of slag. Part II was mainly composed of C, which was a matrix of slag and played a role in assisting melting. Part III was more complex, the main compositions were P1 (Al2O3·4SiO2), P2 (2CaSiO3·CaAl2O4·BaO) and a small amount of P3 (CaF2). Among them, P1 existed alone, while P3 existed around P2”.

The authors are invited to give microanalysis (elements contents) results of these part to prove their composition mentioned above.

Similar comment on this paragraph.

Specifically, Figure 6 was the details of the slag morphology after the slag-steel reaction for 10 minutes. The dendritic crystalline phase was CaF2 which was a benefit to lubrication. The matrix of slag crystallized uniformly along the CaF2 dendritic, and the composition was 3CaO·3SiO2·2Al2O3·BaO. No cuspidine (3CaO·2SiO2·CaF2) with high melting temperature was observed, indicating slag had good performance on uniform thermal transmission and melting characteristic so far.

Similar comments for the description of Figures 7, 8 and 9 with respect to the phases’ composition.

In an alkalescent melting system, according to previous researchers [16, 23],

It seems that Reference [23] is not in the references list. Is this probably Reference [22]?

Furthermore, what is “[J]” in several references?

Hence, the authors are invited to check all References for correctness.

Author Response

Dear Reviewer,

Thanks for your valuable comments. These comments are very useful for our work. Your comments and our responses are in the document "response to reviewer 1"

Reviewer 2 Report

The authors used a low-reactive CaO-SiO2-Al2O3 slag to be used in the casting of Al-80 TRIP steel. A characterization of the slag is presented, followed by a kinetics equilibrium model of SiO2 in the liquid slag was developed to analyse the factors that affect SiO2 consumption.

The manuscript is an interesting but some English corrections are needed. I suggest some other minor changes:

Line 90. Change tense of verb: …is listed in table 1. Samples…

Line 91 and in other places you begin a sentence with And. Delete the dot or delete the preposition “And”.

Explain why you select this two different slags?

Table 1. why you pot t after Al?, this should be deleted.

This is chemical composition and not compositions

Table 2.  This table should be later, after you when explain how you did the slag analyses.

The table caption should be: “ Chemical composition nd properties of…”.

In this table Al should be as Al2O3.

It could be interesting that you calculate Tm not only by the Factsage software but also by Differencial Thermal Analysis (DTA) or differential scanning calorimetry (DSC).

Please, explain how did you obtain the slags.

In the methods section you have to explain all the analytical techniques that you used eg. Scannoing Electron microscopy.

Line 144 in addition to Figure 4 a more extensive description of the morphology of the slags should be indicated inb the text. Shape and size of the different components of parts I, II and III.

You describd a little:

- the morphology of the original slag sample B.

- morphology for slag sample B after slag-steel reaction

We need the same with sample A.

-Line 171 “The dendritic crystalline phase was CaF2

Line 172. “.composition was 3CaO·3SiO2·2Al2O3·BaO. No cuspidine (3CaO·2SiO2·CaF2).. How you do know that?. Did you do XRD?.

The same in the other phase interpretations from here: you need to do XRD.

The section of conclusions must be improved. this is a summary not conclusions.

Author Response

Dear Reviewer,

Thanks for your valuable comments. These comments are very useful for our work. Your comments and our responses are a document "response to reviewer 2"

Reviewer 3 Report

The paper can be recommended to be published after elimination of small errors (see attachment).

Explain the combination of Figure 9 and coefficients a and b in the equations (5) and (6).

Author Response

Dear Reviewer,

Thanks for your valuable comments. These comments are very useful for our work. Your comments and our responses are in the document "response to reviewer 3"

Round  2

Reviewer 1 Report

Please check data corresponding to the evolution of %Na2O (sample B) as a function of time in Figure 2(b). Only one point (initial content of Na2O is obvious in this Figure)

Author Response

Dear reviewer,

Thanks for your suggestion. In fact, we only detected Na2O in the initial composition of the slag but did not find the presence of Na2O in reacted slag. The reason may be, as Kim et al.' s research when the [Al] content in steel is less in 1.8%, the Na2O will be evaporated. In addition, this work focused on the composition change of Al2O3 and SiO2, so the evaporation behavior of Na2O was not emphasized. Nevertheless, we have added relevant content in the manuscript according to your suggestions. 

Best regard,

Kaitian Zhang

Reference: M. S. Kim, S. W. Lee, J. W. Cho, M. S. Park, H. G. Lee and Y. B. Kang. A Reaction Between High Mn-High Al Steel and CaO-SiO2-Type Molten Mold Flux: Part I. Composition Evolution in Molten Mold Flux. Metallurgical and Materials Transactions B, 2013, 44(B): 299-308.

Reviewer 2 Report

I consider that the manuscript was improved, but it still needs some changes:

-The tense of verbs still have not changed, eg. line 103  Figure  1 showed, this should be is present tense.

- In the table captions: You must change chemical compositions by  chemical composition

Author Response

Dear Reviewer,

Thanks for you very much. We have revised the manuscript according to your suggestions.

Best regards

Kaitian Zhang